# Conducting qualitative research on acute mental health inpatient wards: Lessons from the field

**Isobel Johnston**[1], **Helen Morley**[2,1], **Gill Gilworth**[1], **Jessica Raphael**[2], **Paul Wilson**[3], **Dawn Edge**[2,1], **Katherine Berry**[2,1]*

1  Research and Innovation, Rawnsley Building, Manchester Royal Infirmary, Greater Manchester Mental Health NHS Foundation Trust, United Kingdom, 2  Division of Psychology and Mental Health, School of Health Sciences, Zochonis Building, The University of Manchester, United Kingdom, 3  Centre for Primary Care and Health Services Research, School of Health Sciences, Williamson Building, The University of Manchester, United Kingdom

* Katherine.berry@manchester.ac.uk

## Abstract

Qualitative process evaluations incorporating ethnographic observations and semi-structured interviews are increasingly being used to supplement randomised control trials when designing and testing complex health interventions, including in mental health inpatient settings. Yet how these components are conducted is less discussed. In this paper we describe our approach to conducting ethnographic participant observations on acute mental health inpatient wards. We discuss how we mitigated, negotiated, and adapted our research to effectively and sensitively complete these elements. We demonstrate the significance of personal and team reflective practice in guiding the research and provide reflections from the researchers on our direct experience of completing observations. Throughout the paper we discuss how we familiarised ourselves to individual research sites, reflected on our roles on the ward, adapted qualitative research techniques for acute mental health settings and the significance of peer debriefing.

## Introduction

Qualitative process evaluations are increasingly being used to supplement randomised control trials, especially when designing and testing complex health interventions. The qualitative elements provide contextual and environmental understanding [1,2], helping to explain complex causal pathways, facilitators, and barriers to implementation of an intervention in the real-world setting being researched [3,4]. Therefore, they have a clear role in the design and evaluation process for healthcare interventions, including being used in mental health settings.

In health research settings, qualitative methods have been used to evaluate the implementation of a range of interventions. In this paper we focus on research on acute mental health wards. Other examples of qualitative research evaluating the implementation of psychological on acute mental health wards include the evaluation of art therapy groups [5]; talking groups [6]; and training nurses to provide different therapies [7]. Raphael et al (2021) report a meta-synthesis of qualitative studies exploring barriers and facilitators to implementing

**Data availability statement:** Reflective notes taken by researchers conducting qualitative interviews and observations: https://doi.org/10.48420/27844110.v1.

**Funding:** This project was funded by the National Institute for Health Research (NIHR) Programme Grant for Applied Research RP-PG-0216-20009. This study was supported by the NIHR Manchester Biomedical Research Centre (NIHR 203308). The views expressed are those of the authors and not necessarily those of the NIHR or the Department of Health and Social Care.

**Competing interests:** The authors have declared that there are no competing interests.

psychosocial interventions within acute inpatient mental health settings [8]. The authors identified a range of factors influencing implementation under the broad categories of motivation, capability and opportunities. They conclude that motivational barriers can be overcome by good working relationships, provision of clear information, staff inclusion in implementation and intervention design, and good quality staff training. Capability barriers can be overcome by therapist qualities such as empathy, flexible interventions and good staff training. Opportunity barriers can be overcome by strong leadership with core psychological values, team cohesion, staff accountability for delivery of psychosocial interventions and team prioritisation of psychosocial intervention delivery.

Although this qualitative research provides richer insights into the delivery of interventions in practice not captured by quantitative research, studies in acute mental health settings often employ interviews or focus groups to evaluate interventions [8]. As James et al [9] explains: "Whilst interviews and focus groups are well suited to gathering data on participants' views and experiences, they cannot capture enacted behavioural manifestations of intervention delivery" (page 2). Using both interviews and ethnographic observations can increase the researcher's understanding and evaluation of an intervention on acute mental health inpatient wards and observations in particular have been shown to uncover aspects of job roles that are not articulated in interview or questionnaire studies [10]. However, there is limited literature on how to conduct observations and interviews in this context.

In this article we draw on our experiences of conducting ethnographic participant observations and semi-structured interviews as a part a cluster randomised controlled trial exploring the implementation of psychologically informed therapies on acute mental health wards [11]. Throughout the research we employed reflexivity and peer debriefing to guide data collection and analysis. These are both crucial practices in qualitative research and the generation of knowledge through such data collection methods [12–14]. There is limited literature on conducting qualitative research in acute mental health inpatient settings and a clear gap in the literature of how to apply reflexivity and peer debriefing in a qualitative process evaluation of a complex intervention being implemented in this setting.

We begin with a background to the trial, to help situate our reflections in the wider data collection and aims of the research. We then draw on examples from our own reflections to highlight and discuss our qualitative process evaluation research experience from entering to leaving the field and how we collected both interview and observational data. These reflections aid understanding of how to collect data in health service research settings specifically on acute mental health inpatient wards, but also how reflexivity and peer debriefing can be used as important components of qualitative process evaluations.

## Background to the TULIPS Trial and study setting

The TULIPS (Talk, Understand, Listen for In-patient Settings) trial is a cluster randomised controlled trial which aims to improve access to psychological therapy on acute mental health wards (Identifier: NCT03950388, Registered 15th May 2019) [11]. Thirty-four acute mental health wards were randomly assigned to receive the intervention and treatment as usual (the intervention arm) or only treatment as usual (the control arm). The intervention model involved integrating a part-time clinical psychologist into the ward team to facilitate team formulation, reflective practice and train staff to deliver one-to-one and group therapy sessions [11]. Psychologists also delivered psychological therapy on a one-to-one basis to a small case load of service users.

The current article relates to data collection for the qualitative elements of the process evaluation of the main trial, which took place in a sample of study wards in the intervention arm of the trial from December 2019 to July 2022. Our research setting was acute mental health inpatient wards across England. Participating sites were purposely sampled to include a mix

of male and female wards in a large metropolitan city, smaller cities, and towns serving more rural populations.

Three of the authors were involved in the qualitative data collection: GG, JR and IJ. Alongside this was the wider research team KB, DE, HM and PW evaluating the process of the intervention contributed towards the peer debriefing, analysis, and reflection throughout the data collection. Ethnographic observations were non-participant (sitting in on a meeting and only observing) and participant (researchers engaged with the ward environment and participants while observing in communal spaces). Six wards from the intervention arm of the trial were identified for observations. These wards were selected as representative of the wider sample of wards included in the trial in terms of gender composition (four male and two female wards) and location (three rural and three urban). Observations were completed for up to five days at a time and detailed field diary notes were taken on six wards at three time points, pre-, during and post-intervention. The ethnographic observations were used to provide environmental and contextual understanding of what and how the intervention was being implemented on different wards in a mental health setting.

Both purposeful and broader ward observations were completed. Purposeful observations are those relating to specific events, meetings, or therapy sessions on the ward. Including attending and documenting: ward rounds; handover meetings; administration of medication; staff meetings; service user community meetings; and group and individual therapy sessions. Broader observations in public ward spaces were used to capture ward culture and environment outside of these meetings and sessions. Service user participants' and staff members' experiences and views were also investigated via semi-structured interviews. At the time of writing, this included 32 semi-structured interviews with staff, and 31 with service users. Interviewees were recruited both opportunistically and purposively from wards in the intervention arm of the trial. A sampling frame was used to inform purposive sampling. The sampling frames included demographic variables such as gender, ethnicity and age group as well as job role for staff and diagnosis and previous involvement in psychological therapies for patients. Recruitment began relatively opportunistically, but the team met regularly to review participant characteristics and identify characteristics on the frame that were under represented. We then used this information to target the recruitment of individuals who possessed thus far unrepresented characteristics.

When possible and practical, the interviews were carried out in person, however due to COVID-19 some interviews were also held virtually (either using video call or telephone calls).

## Methods

### Ethical considerations

Ethical approval must be sought in preparation for qualitative process evaluations. Our study was approved in July 2019 by the Greater Manchester NHS Research Ethics Committee, IRAS ID: 264686. Consideration of the informed consent process, right to withdraw and minimizing risk should be made and can be informed by multiple available guidelines such as the British Psychological Societies Code of Human Research Ethics [15]. Patient capacity to consent was a particularly important issues to consider in both interviews and observations with patients. Many people on inpatient wards are detained under a section of the Mental Health Act (1983) meaning that they were not deemed to have capacity to consent to treatment for their mental health condition. According to the Mental Health Capacity Act (2005) individual's capacity needs to be assessed in relation to different decisions meaning that people who are deemed to lack of capacity regarding mental health treatment could be perfectly capable to make decisions about participating in research. As in all research, it was nonetheless important that researchers were trained to assess each patient's capacity to consent to the study with support from the clinical team.

Researchers sought written informed consent to attend and observe private meetings and group therapy sessions from all participants present prior to the meeting or session taking place. The acute mental health setting posed additional challenges for the observational study due to the potentially sensitive nature of information discussed during meetings or group therapy sessions. Whilst consent from everyone present was the aim if someone did not consent, they were excluded from the observation notes. Access to areas on the wards was hindered by the ward layout and limited to public spaces potentially reducing privacy for service users. For observations in public spaces posters were displayed to indicate which area researchers were observing and over what time period, so service users or staff could avoid the area if they wished. Transparency about the research, when and where the observations were taking place, gave service users and staff choice and time to consider whether to participate or not in the observations.

Written informed consent, including consent for recording, was obtained prior to all individual interviews. Prior to approaching service users about taking part in interviews or observations the researchers spoke with participants' named nurse or a senior member of the ward-based team to help establish their capacity to provide informed consent. Risk assessments were completed to allow for further consideration of service user wellbeing. Details were sought on signs of distress, communication difficulties and current risk to self or others. Service users were not approached when clinicians did not feel they had capacity to give informed consent or that their participation could have a negative impact on their wellbeing. When gaining consent, the voluntary nature of their participation and right to withdraw were emphasized with researchers ensuring service user participants were aware that their withdrawal or non-participation would not impact their treatment on the ward. Such considerations are of particular importance when collecting data where power dynamics are at play, such as on acute inpatient wards, as there is an enhanced risk of perceived obligation to participate [15].

## Stakeholder Involvement: Public and Patient Involvement

Stakeholder input on studies processes and documentation should also be sought in preparation for, and throughout the lifetime of a study. For example, patient and public involvement (PPI) members with relevant lived experience were recruited for the TULIPS study. The PPI group were consulted throughout the research cycle. Included in the consultation process was the development of interview questions; they suggested amendments to phrasing which were adopted. The PPI group also reviewed participant facing documentation, for example, the participant information sheet (PIS). The members made changes to the wording and presentation of the PIS to make it easier for service users to understand the purpose of the research and what it involved. Finally, the PPI group gave advice on our standard operating procedure in relation to managing risk and distress to participants during the interview process. The central study research team met regularly with the PPI group to provide study updates, so they remained fully informed of developments and could provide feedback.

## Relationship management and initiating the qualitative research

Potential study wards were identified soon after the outcome of randomisation. By this stage research staff from the local site had already been supporting the set-up of the TULIPS study. A flexible, open, and honest approach assisted with relationship building and enabled trust between the research site and central study site team to develop. These established relationships facilitated the introduction of qualitative researchers. It was important for these researchers to establish their own relationship with the ward manager and team including the trial psychologist.

GG: It was meeting the ward manager that helped reassure me there would be the support we needed for the data collection. She helped me build the picture of the ward routine I needed, such as timings for key events I wanted to include [in the observations] such as handover, ward rounds and any groups taking place on the ward.

JR: I started by attending senior management meetings with service managers to present the study to obtain their buy-in. I met with ward managers to understand best times to attend the ward and to obtain their permission to release staff to speak with me and assist with identifying suitable service user-participants.

GG: On one ward the ward manager was shielding because of COVID so I had to meet them on a video call the first time when they were working from home. I think it was good to meet them away from the busy ward environment where there are always lots of interruptions and demands on their time. It felt as if we had more time to build a relationship and for them to take the time to understand what the qualitative work involved.

Taking the time to develop these relationships between the qualitative researcher and key gatekeepers of intervention wards enabled easier access for the latter stages of familiarisation and data collection, better communication with the wider ward team to provide clarity on the researcher's activities and appropriate tailoring of these activities for individual ward contexts.

## Familiarisation and finding a 'place'

After setting up the trial and building relationships with staff at the study wards and sites, the next stage was to build contacts between the researcher and those who would be involved in the research. As with all qualitative research, building positive relationships with participants is vital from the outset, through to leaving the field. Building rapport aids the gathering of in-depth and respectful qualitative data, making it central to both interviewing and ethnographic studies [16–18]. We found that familiarisation between the researchers, and staff and service users on the wards helped with the overall research process in increasing staff buy-in, opportunities to recruit for interview and creating a positive rapport. Spending time meeting ward staff and service users before the qualitative data collection commenced was ideal. The below extracts include JR describing her initial familiarisation with one study ward, and GG discussing how she engaged with staff before the data collection commenced on another.

JR: With ward manager permission, I attended the ward on a frequent basis to familiarise myself with the ward staff, service users and ward routine. I walked around the ward with staff to get familiar with the layout and meet service users and also sat in nursing offices and staff rooms to get familiar with staff.

GG: I attended some of the training that the trial psychologist delivered for the ward staff which gave me an opportunity to explain the qualitative elements of the trial to the staff and gave them an opportunity to ask questions too. When I went on the ward the following week it was good that I had met some of the staff already.

Several familiarisation methods were used to nurture collaboration. Although not dissimilar to other contexts, it is important in healthcare settings and particularly on acute inpatient wards, due to the uniqueness of the environment, that all suitable familiarisation strategies are employed. Acute mental health wards often present discreet and 'close-knit' staff cultures [19] making them

harder to penetrate as an outside observer in comparison to other healthcare settings. Due to service users' familiarity with wards (being inpatients) they were aware of who staff were and therefore, were more likely to take notice and be wary off outside researchers. As a result, observers were often approached by service users wanting to clarify who they were and what they were doing.

> IJ: When spending time sat in communal areas service users would often approach me and ask if I was a student nurse and what I was taking notes on. Others would sometimes walk by and look towards me with what felt like wariness or suspicion. I found after introducing myself and the study to these individuals they would pay less attention to my presence when moving around the ward, or, would actively engage with me and discuss their views on the ward.

Anxieties and paranoia were also present among staff with concerns over external scrutiny being heightened due to current pressures to reduce restrictive practices on mental health wards in general [20,21]. These are but a few examples of the discreet nature of acute mental health wards which make the familiarisation stage, and the use of participant observations, pertinent to allow for participants to acclimatise to the researcher's presence for a more accurate picture of the ward to be gathered.

Observers interacted with the ward environment, particularly when in communal spaces, through engaging in conversations with service users and staff. This acted to further reduce suspicion around the researcher's presence, increase the richness of field notes and provided opportunities to recruit for interviews. Although observers did not actively participate in the running of the ward, care provided to service users, or the implementation of interventions clarity on their presence, and interactions with the ward environment allowed observers to better integrate into the ward. We found the more researchers could observe the interventions in practice as if they were part of the ward and its culture, the more we could understand how, when, and why they were, or were not, being implemented.

Familiarisation methods included shadowing a member of staff on the ward, with staff members introducing the researcher to colleagues or service users. In some instances, assisting in quantitative data collection offered the opportunity to start the familiarisation process and meet staff and service users in advance of observations. Therefore, relationships and participant familiarity with the researcher could be built prior to the qualitative data collection.

During the familiarisation stage, we reflected on and observed the ward environment itself. This included noting where there were spaces to conduct interviews, where the staffroom and communal service user areas were. We also made notes on ward routines including when handovers and other staff meetings occurred, how shift patterns were organised, where the alarms were, protected time for meals, the timing of medication rounds and regular ward activities. With medication rounds, for example, we reflected that it was often not appropriate to arrange an interview with staff or service users around this time. Medication rounds took a staff member off the floor and therefore removing another for an interview could leave already short staffed wards in a difficult position, which could have implications for staff and service user wellbeing. Side effects of medications, such as drowsiness, meant that service users often asked for interviews to be a few hours after taking their medication. Observing these small details during familiarisation meant that we could be respectful of the daily ward routine during qualitative data collection, whilst spending time with staff and service users also helped us to become an accepted, embodied presence in the everyday of those within the field site [22].

An important point of familiarisation with the wards was to attend handovers to gain current information about service users. Handovers are a meeting between staff prior to a shift change to provide staff coming on shift with updates, discuss service user wellbeing during a previous shift and actions that need to be implemented during the next shift. For instance, we

heard which service users may be suitable to be approached for an interview or which service users were not having a "good day" and therefore might be better not approached that day. Additionally, ward managers would provide a daily update on service user acuity, ward stability, and staffing levels which all impact on whether we may be able to conduct interviews.

Throughout this period, we also reflected on how to create a 'place' for ourselves within the field site (acute inpatient ward) [22], one which would balance being accepted by participants, causing minimal disruption to their daily activities and the wider ward environment, whilst facilitating data collection. It was especially significant for the ethnographic observations, which involved prolonged periods of time spent on the wards, to observe barriers and facilitators to the intervention implementation in different spaces. It was evident our presence influenced people's behaviours, with staff members, for instance, sometimes muting their conversation to each other when a researcher passed by. We observed how anxieties or suspicion towards the researcher were slowly reduced through spending more time on the wards and participants becoming acquainted with the researcher.

## Defining our role on the ward

Entering the field site a researcher begins a performance of self, with the aim of conveying an impression which will, at its core, benefit the data collection [23]. The nature of this ward-based qualitative data collection meant that there were prolonged interactions with participants, which took the form of both informal interactions and those relating to the research (e.g. conducting interviews). This resulted in the continual need to be reflective on our presentation as a researcher on the ward, and our role within the wards, including indicating this through non-linguistic communication.

Clothing is well known to be a marker of inclusion or exclusion within social groups [24], thus it is a significant factor to consider in interactional research [16]. Clothing was particularly significant when observations or interviews were being held on wards where staff members wore uniforms. On mental health inpatient wards clothing defined the role of an individual. Ward staff wore uniforms of different colours and styles which denoted their role. Doctors did not usually wear a uniform although during the Covid-19 pandemic they did revert to wearing scrubs.

> JR: Based on my own reflection of working in hospitalised settings I felt it was important to 'dress down' when going on to the wards so to not come across as a senior professional and in which case put off staff (given the hierarchical nature of the ward). By dressing in a casual way, I also hoped to address power-imbalances with service users. I hoped to portray less of an 'us and them' feeling with service users, given anecdotal information regarding lack of service user trust in staff and hierarchies, so they could see me as someone who would identify with them more so than a staff member and therefore hope that they would be more willing to engage in the research. Interestingly whilst on the ward service users asked both whether I was another service user on the ward, or a doctor.

In these instances, different pieces of clothing were emblems of a group on the ward. Casual clothing was interpreted as being a service user, whereas black trousers and a lanyard was a symbol of being a doctor. Whilst we made attempts to reflect on and adapt our clothing to create the correct impression moments of misunderstanding between the researcher and individuals on the ward still occurred, and so potential frustration towards the researcher for not fulfilling the role participants expected. Therefore, we reflected that finding our place on the ward also involved creating a clear understanding that we were researchers whenever speaking to someone for the first time, or if they had forgotten who we were. Through doing this, we could mitigate misinterpretations about our role.

During peer debriefing we continually reflected on the roles and actions we were taking on the wards, with the agreed principle that helping in small ways for example opening the door for service users, was beneficial to the research, researcher, and participants.

## Data collection

As described above there were two qualitative elements for the process evaluation used, semi-structured interviews and ethnographic observation. In this section we discuss reflections on conducting each of these, how we adapted the research techniques, and mitigated challenges faced in the qualitative data collection.

### Conducting interviews

There were several challenges in conducting interviews on acute inpatient wards, in this section we highlight a selection of these and how we overcame them. The first related to short staffing and the acuity of service users' mental health needs on most of the wards taking part in the TULIPS trial. We frequently experienced walking onto a ward and being told by a member of staff that they were short staffed that day, which is a common occurrence on acute wards [25]. Therefore, completing an interview with a member of staff arranged on a certain day, at specific time, rarely occurred due to a lack of staff time. For example, on one ward GG arrived one morning to find that there were three service users who needed one-to-one observations so three staff were allocated to complete that work, leaving the remaining staff stretched.

Even when wards were adequately staffed, the schedule could frequently change. This could mean prolonged periods waiting for staff to become avaliable. Throughout the project we increasingly used telephone interviews or video calls for interviews with staff participants, which was both more efficient for the researchers' time – in that if it was cancelled, they had not travelled to the ward – and more flexible for staff. Some staff agreed to be interviewed over the phone on their day off if it was proving difficult to fit in during a busy shift.

> GG: There were multiple occasions when I arranged a Microsoft Teams meeting with one of the deputy ward managers to interview her and she either cancelled a few hours before or just didn't log on for the meeting at the time we'd arranged. It was always because they were short of staff, or the ward was too busy that day. That happened every week for about 6 weeks. On the other hand, the ward manager on another ward e-mailed me in the morning to see if I could do the video call for the interview an hour earlier than we had arranged as something else had come up that she had to go to. Video calls are easier in that respect that you can change the time and work around the participant's time more easily.

There were, however, sometimes drawbacks to remote interviewing including the need for a good phone signal or internet connection. Absence of visual cues with telephone or audio only interviews could also interrupt the flow of conversation and removed a further source of information about the interviewees' views and experiences.

The dynamic nature of the wards meant that organised ward activities were prone to change. As such service users, rightly so, attended the activity in place of participating in an interview. Although service users' ward rounds were booked for certain times, they could also be changed at short notice. For example, we had service users who consented to the researcher being in their ward round. However, these were sometimes changed to a time when the research was not present. Therefore, we were unable to attend.

Ward rounds can be difficult times for service users [26], where they may hear about their section being extended, medication changes or discharge. Therefore, understandably, if an

interview had been arranged some service users preferred to move it to another day or time. They may also be granted leave in their ward round, and so took the opportunity to leave the ward immediately afterwards. Researchers were often unaware of events or ward rounds moving until they had reached the ward. Again, this meant that flexibility was needed whilst conducting data collection, to work around participant's time and ward life, to take a respectful approach to the research. This could be through arranging another time to conduct the interview, offering to do a virtual interview instead or finding spaces on the ward to complete other work allowing more flexibility in when interviews were completed.

> GG: I'd arranged to interview 'S' in the afternoon of one of my observation days but when she had ward round that morning 'S' found out she could go home that afternoon. When I went to her room 'S' was busy packing and trying to get hold of her dad on the phone to arrange a lift home. She was really excited, and I could see it was not an appropriate time to expect her to concentrate on speaking with me. 'S' agreed I could ring her a couple of days later, so we did the interview over the phone once she was home.

Clearly there are differences between interviewing a service user participant on the ward compared to after discharge once they are home. Arguably most service users are probably more relaxed in their home environment compared to on an acute ward. However, all interviews were preceded by a risk assessment which could be challenging to complete for people once they had been discharged. On the ward it was relatively easy for the researcher to speak with the participants named nurse to complete this. Once someone was home contacting the individual's care coordinator or appropriate community staff could be challenging and time consuming for researchers. Gaining written consent for service user participants who completed remote interviews was also logistically challenging.

In instances where interviews were to be conducted once a service user had been discharged these were generally arranged as phone or video calls but clearly this then relied on the service user understanding the need for a quiet space to help them to concentrate and for recording purposes. Wherever possible this would be explained to the service user face-to-face, like in the example given above for the interview with 'S'. Flexibility was needed to ensure that interviews occurred when service users were, as one service user participant said, "in a good place" to participate in the interview. The research had to be led by the participants, which required flexibility from the researcher to work around how the service user was feeling and their schedule. This could be resource and time intensive:

> GG: I had confirmed by text what time I would ring one lady having already spoken to her on the phone a couple of days before about what the interview would involve and answering her questions about it. But when I rang there was quite a bit of background noise, so I asked if it was still convenient to do the interview. She said yes it would be fine although she should let me know that she was on the bus. I had to say to her I didn't think it would work too well if she was on the bus, so I arranged to ring her again the following day. In fact, it took a few more attempts before I caught her at a good time to do the interview because the next time, I rang she was very sleepy.

On the ward the need to find an appropriate, quiet and safe space to conduct qualitative interviews could also be an issue. Space was typically limited, over-exposed (e.g. windows in place of walls) and regularly interrupted by alarms, staff walking into the room, or distracting noises on the ward. Some wards had a quiet room or family room (a space where service users could see visitors or meet with healthcare professionals from off the ward such as social workers).

Trying to establish when a room was available could be challenging and change at short notice which could mean delays to the interview. Some service users found it frustrating, that we had arranged a time to do an interview but there was no appropriate space to go to. This could be another advantage of completing interviews by phone or video call particularly on wards where service users had a single bedroom that they could use to take a call.

## Observational study

The ethnographic observational part of the study also presented several challenges. The physical layout of most wards meant that the public spaces where researchers spent most of their time were confined to the corridors, lounge or common room, dining area (this was off the ward in some instances) or activity rooms, potentially limiting the accessibility to staff and service user interactions. Service user's bedroom or bed spaces (some of the wards still had shared 4-bed bays) were respected as private unless service users specifically invited a researcher into that space. If the nurses were busy in the office there were periods where there was little ward activity and few interactions between staff and service users to observe.

> GG: I was aware that although the nurses were busy, I was sometimes hanging around in the corridor or lounge without a lot to see or do. A lot of service users get up late in the mornings, so I noticed this more on an early shift, I think service users and staff sometimes wondered what I was doing there even though I tried to explain.

Researchers also spent time in the nurses' office but this room was often crowded and busy so there was sometimes an unspoken perception of the researcher being in the way. Occasionally this was more overt:

> GG: On one ward I had spent a lot of time out on the ward, so I went into the nurses' office where there were quite a few staff already. One of the staff nurses asked me abruptly "do you want something"? I replied "nothing in particular"; she just sighed and carried on with what she was doing; she seemed uncomfortable with me being in the office.

However, spending time in the office was a good opportunity to observe interaction between staff and to learn about the relationship between the trial psychologist and the ward staff.

> IJ: I noticed that as the office was a staff only space it was often used as a place to get respite from the ward, particularly on busier days. Outside of formal meetings it was also the prime point of contact for Trial psychologists and nursing staff. I regular observed the psychologist popping into the office to say good morning on all observation wards which could often develop into an informal formulation or supervision session.

Ward staff were usually welcoming and enthusiastic about the research once the project had been fully explained to them, particularly the purpose of the observations as part of the process evaluation, putting them at ease. However, the example given by GG above highlights how the researchers had to carefully reflect on the research process, and the significance of openness about the research purpose. This is related to another issue which may be best described as inaccurate interpretation of purpose. Some staff were also cautious of researchers initially due to a lack of clarity on blinding protocol.

> IJ: On one ward nursing staff were very distant and often avoided talking to me or making eye contact. After I mentioned the psychologist to a member of staff they were surprised and asked

how I knew them. It became apparent that staff had assumed I was blinded to the research and had considerable anxiety around unblinding other research staff who attend the ward. After this was clarified they were significantly warmer to my presence and more open to engaging.

On the same ward where the incident in the nurse's office occurred an advanced practitioner verbally consented at the time to GG observing a ward round she was leading for a service user who had been keen to help with the research. However, when GG spoke to the advanced practitioner afterwards, they said that they felt uncomfortable about it and did not give written consent for the field notes relating to the ward round to be included in the data collection. The TULIPS trial psychologist working on that ward confirmed that some of the nursing staff were uncomfortable with the observation days. The following is an extract from an email sent by a trial psychologist to GG, when GG asked about how staff felt towards the observations.

> Trial psychologist: "It's obviously something they're not used to and worry they are being assessed or judged in some way even though that's not the case".

Following discussion with the psychologist and within the research team, we concluded we should spend more time reassuring staff, alongside service users, that the purpose of the observational work was to improve the intervention and understand how and when it was being used. Normalising that people may find it uncomfortable when they know observational work is being undertaken. There were also incidences of gatekeeping researchers' access to meetings, particularly by professionals who had a lack of awareness/exposure to the trial psychologist, or, were openly resistant to psychological input on the ward.

We followed this same practice to reassure service users and explain the purpose of the research and what the research involved. There could be initial concern around the role of the researcher, for instance one service user thought a researcher was a journalist and would be publishing an article about what they saw take place on the ward. We ensured that we continually explained the research to staff and service users, reassuring them about anonymity and confidentiality of the research findings. We explained, alongside having research posters on the wards, that participants could tell the researcher that they did not wish to be part of the observations. Once individuals understood the reasons behind the project, they were often happy to discuss their feelings towards psychologists and the ward environment and their ideas about how this could be improved. Explaining the research in-depth either opened a space for participants to express their views or gave them the space to explain that they did not wish to be part of the research. The only exception in relation to confidentiality was if any safeguarding concerns were raised during the observations and/or interviews. Clearly any information of this nature must be disclosed and acted upon. There were other examples where service users seemed to think that they could use the presence of the researcher to help with problems that they were experiencing on the ward:

> GG: I had been chatting to T (a male service user) about the research and explained the observational work I was doing on the ward. T was very critical of the ward and told me that he was planning to write to various bodies to complain. T was obsessed with food and what he saw as the wards inability to cater for his dietary needs. At mealtimes when he had conversations with staff about food afterwards, he would say things like "did you get that? I hope that you wrote all that down"?

## Debriefing

Throughout the data collection we used debriefing for two purposes: to reflect on the research findings and data collection; and to maintain researcher wellbeing. In qualitative research,

peer debriefing enhances the trustworthiness of the data collection, through peers providing critical feedback on the findings [14]. Our research team met regularly throughout data collection to discuss the emerging findings, and critically evaluate ethnographic field notes.

## Debriefing for the research

Peer debriefing was central for the qualitative process evaluation and feeding findings back into the intervention design and delivery. Monthly meetings with the qualitative team, alongside individual meetings between the researcher and their clinical psychologist supervisor were used as a time to feedback on research findings and discuss the data collection. Notes were kept throughout the meetings as an audit trail, and to provide material to reflect on throughout the data collection, ensuring that the conversations turned into direct actions in the research field. For example, following discussions on some of the nursing staff being uncomfortable with the observation days more importance was placed on ensuring staff were aware that the role of the researcher on the ward was not to assess or judge their actions. In preparatory meetings and ward visits to new wards participating in this part of the process evaluation and during subsequent observational work researchers emphasised to staff that it was normal for them to feel strange and possibly even uncomfortable having someone on the ward watching what was going on.

Peer debriefing also helped to question, reflect and address research bias. It creates an opportunity for the researcher and wider team to think about how their positionality may impact on the data collection and analysis. As with many qualitative process evaluations or larger research studies, this trial had several qualitative data collectors, each with a different disciplinary background. Positionality centres on the notion that: "We are differently positioned subjects with different biographies; we are not dematerialized, disembodied entities" [13] (page 248). Many of the researchers involved in data collection and analysis had experience working in inpatient mental health settings. Researchers therefore already held understandings of ward cultures, processes and the local language used, providing context to interviewees' experiences. This background influenced interpretations with researchers being more likely to focus on salient features of their own ward experiences, such as wards lacking resources and conflict within staff-patient relationships. Therefore, members of the wider qualitative research team read the ethnographic notes to also help guide the researchers and discuss how their backgrounds may influence what they were observing or asking about.

## Debriefing for the researcher

Peer debriefing for the researcher, focused on their own mental wellbeing, is often overlooked in research. It may be overlooked due to lack of time and a failure to make time due to researchers' potential discomfort in discussing emotions and limited capacity to mentalise. Despite these potential limitations of reflective practice, we viewed researcher debriefing as fundamental. Although researchers may step off the ward and outside of the field site, their experiences do not always get left at the door. Research has indicated that conducting qualitative research can lead to distress, anxiety, feelings of isolation, emotional upset and intense fatigue among researchers [27]. This is a lesser discussed area in research but a particularly important one when research is conducted on sensitive topics [28]. Providing peer debriefing and supervision focused on the impact of the research on the researcher was an important stage in protecting researchers' wellbeing and the integrity of the data collected.

Although these risks come hand in hand with collecting high quality qualitative data in the context of mental health wards, we propose that these risks can be mitigated by implementing interventions, including but not limited to those employed within this study (regular supervision, peer debriefing and clear standard operating procedures around ending observations due

to heightened risks). We suggest that such processes should be considered and implemented when planning qualitative research in such settings.

## Results

The approaches discussed and employed within the TULIPS trial enabled the successful collection and ongoing analysis of rich qualitative data for the studies process evaluation. Specifically, 32 staff members and 31 service users from intervention wards completed semi-structured interviews providing in-depth insights into the enablers and barriers to implementing the studies interventions, the mechanisms by which the intervention worked and outcomes.

Detailed ethnographic observation notes were collected at three time points across five observation wards, and at two time points on the sixth ward (the third time point could not be collected due to covid-19 restrictions). This totalled at 411 hours and 25 minutes of observations collected across 64 observation days (Averaging 6 hrs and 26 minutes per day, and a mean average of 4 observation days per time point). A total of 166 participants were approached to consent to being observed during private meetings with 157 of these providing consent (45 service users and 112 staff participants). Meetings and other specified activities that were observed included: ward rounds; one-to-one therapy sessions; group therapy sessions; formulations; staff meetings; activity clubs; medication administration; handover. The total number of individuals included in observations in communal areas cannot be calculated as these individuals were not required to provide informed consent.

We propose that data collected within qualitative process evaluations be analysed first individually and then triangulated across sources to allow for comparisons to be made. For example, in the TULIPS study staff and service user interviews, and ethnographic observation notes were qualitatively analysed individually using thematic analysis supported by Nvivo 12 software. The emergent themes from each data source were then triangulated using the framework matrices function to allow for themes to be directly compared. This added an additional layer of complexity to our findings by considering areas where themes aligned, and did not, across data sources, as well as themes which were unique to one source.

This approach to data collection and analysis also allows for opportunities for additional analyses such as case comparisons. Within TULIPS we aimed to compare interviewees experiences between wards based on factors identified both in the observation and fidelity data (e.g. on the basis of the successful vs unsuccessful implementation of interventions between wards). This may be particularly helpful when considering the impact of multiple complex variables such as ward culture, on the implementation, engagement and resulting outcomes of a complex intervention, such as in the TULIPS trial.

## Discussion

This article has provided guidance in the aim of assisting future researchers in the collection and analysis of data for qualitative process evaluations. Drawing on our reflections from the TULIPS study we have provided specific guidance and considerations when preparing, collecting and analysing data from semi-structured Interviews and ethnographic observations. Although much of this guidance is relevant across health research, we offer multiple suggestions specific to data collection in inpatient mental health settings, an area which is otherwise lacking within the methodological literature. Further implications of this paper for future research and conclusions on its importance within the field will be discussed.

### Implications for future research

One key outcome of this study is the implication of these reflections for researchers conducting qualitative process evaluations within health care settings, and more specifically, acute inpatient wards. We propose that researchers should spend extended periods in the environment in question to enable familiarisation with the context and participants of interest. This process can aid researchers in embedding themselves within the observation context, reducing suspicion and behavioural censoring. Given the often close-knit teams on psychiatric wards, investing time to engage with staff and service users can assist in finding a place and reducing paranoia, particularly among patients, which can increase the richness and accuracy of data collected. It is key to include more general engagements with the environment, staff and patients, not only those required as a part of the implementation of interventions. Psychiatric wards can be highly changeable and understaffed environments, limiting staff's engagement with qualitative research which can be time consuming. To successfully collect qualitative data in these settings researchers must be extremely time flexible. Providing the opportunity to complete interviews remotely, or outside of working hours can be beneficial. Spending prolonged periods on wards to offer greater opportunities to engage interviewees when they are willing and able to participate. Considerations for reflexivity and the inclusion of peer debriefing can enhance support for researchers and add credibility to findings. Reflecting both on the research process as well as researchers approach to collection and analysis, can enable the research team to explore ways to enhance data collection.

In summary, qualitative research within health care settings can provide valuable information about factors influencing the implementation and delivery of interventions during randomised control trials. However, health care settings are associated with unique challenges to conducting qualitative research which are difficult to navigate. Whilst some challenges are likely to be relevant across a range of different settings, others are specific to the area of health care being studied. It is important for researchers to write about their experiences of the processes of conducting research within each environment so that others can learn from these experiences. We therefore believe that this paper provides an invaluable resource for others embarking on research within acute mental health settings. This paper also provides an important template or point of comparison for those who want to describe qualitative research within other health care settings and more specifically within acute inpatient wards.

It is likely that other specialist health care settings will pose additional or different challenges to conducting qualitative research. Further exploration in such environments, for example forensic psychiatric settings, would be beneficial in providing guidance to future researchers aiming to conduct qualitative process evaluations in these settings.

### Author contributions

**Conceptualization:** Isobel Johnston, Jessica Raphael, Katherine Berry.

**Formal analysis:** Isobel Johnston.

**Supervision:** Katherine Berry.

**Writing – original draft:** Isobel Johnston.

**Writing – review & editing:** Helen Morley, Gill Gilworth, Jessica Raphael, Paul Wilson, Dawn Edge, Katherine Berry.

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
