## [Decision Letter · Decision Letter 0]

8 Aug 2024

PONE-D-24-24720Conducting Qualitative Research on Acute Mental Health Inpatient Wards: Lessons from the FieldPLOS ONE

Dear Dr. Berry,,

Thank you for submitting your manuscript to PLOS ONE. After careful consideration, we feel that it has merit but does not fully meet PLOS ONE’s publication criteria as it currently stands. Therefore, we invite you to submit a revised version of the manuscript that addresses the points raised during the review process.

The study provides important insight into the subject. However, a thorough review is needed to improve the syntax and grammar of the manuscript. ==============================

We look forward to receiving your revised manuscript.

Kind regards,

Shazia Khalid, PhD

Academic Editor

PLOS ONE

2. Please ensure you have included the registration number for the clinical trial referenced in the manuscript.

 [The TULIPs Trial was funded by the National Institute of Health Research (NIHR) RP-PG-0216-20009. ].  

5. We note that your Data Availability Statement is currently as follows: [All relevant data are within the manuscript and its supporting information]

Reviewers' comments:

Reviewer's Responses to Questions

**Comments to the Author**

1. Is the manuscript technically sound, and do the data support the conclusions?

Reviewer #1: Yes

Reviewer #2: Yes

2. Has the statistical analysis been performed appropriately and rigorously? 

Reviewer #1: No

Reviewer #2: N/A

3. Have the authors made all data underlying the findings in their manuscript fully available?

Reviewer #1: Yes

Reviewer #2: No

4. Is the manuscript presented in an intelligible fashion and written in standard English?

Reviewer #1: Yes

Reviewer #2: Yes

5. Review Comments to the Author

Reviewer #1: The current paper presents a valuable contribution to the field of mental health research by providing detailed methodological insights and practical reflections on conducting qualitative research in acute mental health in-patient wards. The article addresses a significant gap in the literature on how to conduct qualitative process evaluations which is its biggest strength. Paper also have methodological rigour, the detailed description of ethnographic observations and semi-structured interviews, along with the challenges faced and strategies employed, demonstrates a robust and reflective approach to qualitative research. By incorporating some of the revisions to enhance analytical depth, structural clarity, and integration with existing literature, the paper would be well-suited for publication and could serve as a significant source for researchers in similar fields.

• In line 4 & 5 of Abstract, words “including in mental health care settings” should be replaced with “including mental health in-patient settings” so it may go with the title of the study.

• Incorporate a more extensive review of related literature to contextualise the study’s methodology and findings

• The sampling strategy needs to be explained, how hospitals were selected, how sample and wards were selected, which guidelines were followed.

• Specify clearly about selection of sample by using tables or flow charts so that the reader may understand it more easily

• Observation guide and interview guide should be included will clear cut points

• Reorganize sections with clearer subheadings and ensure a logical flow from one section to the next to enhance readability and comprehension. The paper could be improved with a clearer structure and flow. Some sections, such as the methodology and results, might benefit from more subheadings or a clearer delineation of subtopics to enhance readability.

• The word “such as” is used many times which shows that only some of the data is shared not all. This shouldn’t be the style to report rather include all in the form of tables or flow charts. E.g. Line no 84, 106, 139, 140, 216, 234, 320, 508.

• Use of randomization should be included in detail about which the researcher wrote in line 130

• Results of the interviews and observational study must be included in detail with tables and flow charts regarding themes generated which make sense and understanding to the reader. Provide a more thorough analysis linking specific observations to theoretical frameworks or broader themes in qualitative and mental health research.

Reviewer #2: Overall, the paper “Conducting Qualitative Research on Acute Mental Health Inpatient Wards: Lessons from the Field” provides a significant foundation for understanding the use of qualitative process evaluations in mental health care settings. The paper discusses how qualitative research techniques were adapted for acute mental health settings. This specificity is crucial for ensuring that the methods are appropriate and effective in the given context.

The use of both ethnographic observations and semi-structured interviews to supplement randomized control trials in mental health care settings is credible.

The paper offers a clear description of the approach to conducting ethnographic participant observations on acute mental health in-patient wards.

Highlighting the significance of personal and team reflective practice adds depth to the research process. It shows an awareness of the researchers' impact on the study and the importance of continuous self-evaluation.

This sis a review paper which may offer even greater contributions to the field by addressing the identified issues.

Limitations of the Research Paper

The long sentences are used in the document (line 56-59 and 65-68) it’s better to make simple and small sentence so that the reader may get benefit from the paper.

The paper doesn't talk much about the specific problems faced during the ethnographic observation part of the study, which could have offered helpful insights into practical difficulties.

Although the paper extensively covers the importance of personal and team reflective practices, it doesn't go deep into the possible limitations of this approach, missing a critical evaluation of its effectiveness.

There's limited exploration of potential biases that might have influenced the researchers' observations and interpretations, which could affect the study's validity and reliability.

The paper doesn't address how the researchers' backgrounds, experiences, and perspectives might have impacted data collection and analysis, which could affect the generalizability of the results.

The paper could benefit from a more detailed discussion on the ethical considerations and challenges encountered during the research, especially in sensitive settings like acute mental health wards.

The paper could have included a more detailed discussion on the study's implications for future research and practice in acute mental health settings, highlighting areas for further investigation and development.

6. PLOS authors have the option to publish the peer review history of their article (what does this mean?). If published, this will include your full peer review and any attached files.

Reviewer #1: No

Reviewer #2: No

---

## [Author Response · Author response to Decision Letter 1]

11 Dec 2024

Dear Editor

Thank you for the opportunity to submit a revision of our manuscript “Conducting Qualitative Research on Acute Mental Health Inpatient Wards: Lessons from the Field”. Please find the reviewer and editor comments below in bold, followed by a description of how we have addressed these comments and changes made to the manuscript in italics where relevant.

Journal Requirements:

2. Please ensure you have included the registration number for the clinical trial referenced in the manuscript.

This is now included in the ‘Background to the TULIPS Trial and Study Setting’ section on page 4.

(Identifier: NCT03950388, Registered 15th May 2019)

We have added the following under funding information

This project was funded by the National Institute for Health Research (NIHR) Programme Grant for Applied Research RP-PG-0216-20009. This study was supported by the NIHR Manchester Biomedical Research Centre (NIHR 203308). The views expressed are those of the authors and not necessarily those of the NIHR or the Department of Health and Social Care.

4. Please state what role the funders took in the study.

The funders had no role in study design, data collection and analysis, decision to publish, or preparation of the manuscript. This has been updated in the Financial information section (see response to previous point).

5. We note that your Data Availability Statement is currently as follows: [All relevant data are within the manuscript and its supporting information]

Please confirm at this time whether or not your submission contains all raw data required to replicate the results of your study. Authors must share the “minimal data set” for their submission. PLOS defines the minimal data set to consist of the data required to replicate all study findings reported in the article, as well as related metadata and methods (https://journals.plos.org/plosone/s/data-availability#loc-minimal-data-set-definition [journals.plos.org]).

If your submission does not contain these data, please either upload them as Supporting Information files or deposit them to a stable, public repository and provide us with the relevant URLs, DOIs, or accession numbers. For a list of recommended repositories, please see https://journals.plos.org/plosone/s/recommended-repositories [journals.plos.org].

We have included a repository of researchers’ reflections from field work and interviews which informed this paper. This repository can be found at https://doi.org/10.48420/27844110.v1 and is included in text on lines 679-681::

Supporting Information

Reflective notes taken by researchers conducting qualitative interviews and observations: https://doi.org/10.48420/27844110.v1

6.Your ethics statement should only appear in the Methods section of your manuscript. If your ethics statement is written in any section besides the Methods, please move it to the Methods section and delete it from any other section. Please ensure that your ethics statement is included in your manuscript, as the ethics statement entered into the online submission form will not be published alongside your manuscript.

We can confirm that the ethics statement is now within the methods section of the paper.

We can confirm that all included references have been checked for accuracy and are cited within the text.

Reviewer Comments:

1. In line 4 & 5 of Abstract, words “including in mental health care settings” should be replaced with “including mental health in-patient settings” so it may go with the title of the study.

Updated in text with including mental health in-patient settings.

2. Incorporate a more extensive review of related literature to contextualise the study’s methodology and findings

We have added the following relating to qualitative studies in inpatient settings

Other examples of qualitative research evaluating the implementation of psychological on acute mental health wards include the evaluation of art therapy groups [5]; talking groups [6]; and training nurses to provide different therapies [7]. Raphael et al (2021) report a meta-synthesis of qualitative studies exploring barriers and facilitators to implementing psychosocial interventions within acute inpatient mental health settings[8]. The authors identified a range of influences influencing implementation under the broad categories of motivation, capability and opportunities. They conclude that motivational barriers can be overcome by good working relationships, provision of clear information, staff inclusion in implementation and intervention design and good quality staff training. Capability barriers can be overcome by therapist qualities such as empathy, flexible interventions and good staff training. Opportunity barriers can be overcome by strong leadership with core psychological values, team cohesion, staff accountability for delivery of psychosocial interventions and team prioritisation of psychosocial intervention delivery. Although this qualitative research provides richer insights into the delivery of interventions in practice not captured by quantitative research, studies in acute mental health settings often evaluate an intervention using interviews or focus groups [8].

Using both interviews and ethnographic observations can increase the researcher’s understanding and evaluation of an intervention on acute mental health in-patient wards and observations in particular have been shown to undercover aspects of job roles that are not articulated in interview or questionnaire studies[10]

3. The sampling strategy needs to be explained, how hospitals were selected, how sample and wards were selected, which guidelines were followed.

Additional details have been added in text.

Six wards from the intervention arm of the trial were identified for observations. These wards were selected as representative of the wider sample of wards included in the trial in terms of gender composition (four male and two female wards) and location (three rural and three urban).

4. Specify clearly about selection of sample by using tables or flow charts so that the reader may understand it more easily

We have added the following text but feel that a table of the sampling frame or flow charts would be too prescriptive given that our method was both opportunistic and purposeful.

Interviewees were recruited both opportunistically and purposively from wards in the intervention arm of the trial. A sampling frame was used to inform purposive sampling. The sampling frames included demographic variables such as gender, ethnicity and age group as well as job role for staff and diagnosis and previous involvement in psychological therapies for patients. Recruitment began relatively opportunistically, but the team met regularly to review participant characteristics and identify characteristics on the frame that were under represented. We then used this information to target the recruitment of individuals who possessed thus far unrepresented characteristics.

5. Observation guide and interview guide should be included will clear cut points

We have included a synopsis of the guides in supplementary material

6. Reorganize sections with clearer subheadings and ensure a logical flow from one section to the next to enhance readability and comprehension. The paper could be improved with a clearer structure and flow. Some sections, such as the methodology and results, might benefit from more subheadings or a clearer delineation of subtopics to enhance readability.

We have organised the paper using a more traditional format of method, results and discussion. We note our paper is dedicated to a detailed description of a research methodology so the methods section is relatively long but we felt that the more traditional paper format would help to structure the paper and also address the reviewer’s concerns about subheadings and flow.

7. The word “such as” is used many times which shows that only some of the data is shared not all. This shouldn’t be the style to report rather include all in the form of tables or flow charts. E.g. Line no 84, 106, 139, 140, 216, 234, 320, 508.

At points this phrase has been used to provide examples of which there are a multitude (e.g. each ward offered different activities). Where possible to include these in a succinct way we have added additional details, and at other points this phrase has been removed. Some examples are also derived from exact quotes from researchers’ reflective journals so cannot be edited (e.g. line no 139, 140).

Broader observations in public ward spaces were used to capture ward culture and environment outside of these meetings and sessions.

Risk assessments were completed to allow for further consideration of service user wellbeing. Details were sought on signs of distress, communication difficulties and current risk to self or others.

We also made notes on ward routines including when handovers and other staff meetings occurred, how shift patterns were organised, where the alarms were, protected time for meals, the timings of medication rounds and regular ward activities.

Additionally, ward managers would provide a daily update on service user acuity, ward stability, and staffing levels which all impact on whether we may be able to conduct interviews.

The dynamic nature of the wards meant that organised ward activities were prone to change.

Research has indicated that conducting qualitative research can lead to distress, anxiety, feeling of isolation, emotional upset and intense fatigue among researchers [26].

8. Use of randomization should be included in detail about which the researcher wrote in line 130

This has been updated in the methods section of the manuscript.

The TULIPS (Talk, Understand, Listen for In-patient Settings) trial is a cluster randomised controlled trial which aims to improve access to psychological therapy on acute mental health wards (Identifier: NCT03950388, Registered 15th May 2019) [10]. Thirty-four acute mental health wards were randomly assigned to receive the intervention and treatment as usual (the intervention arm) or only treatment as usual (the control arm).

9. Results of the interviews and observational study must be included in detail with tables and flow charts regarding themes generated which make sense and understanding to the reader. Provide a more thorough analysis linking specific observations to theoretical frameworks or broader themes in qualitative and mental health research.

An in-depth analysis of interview data and observational data is reported in two papers currently in submission. The current article aimed to present reflections on the process of conducting both interviews and observations in this setting, not to provide an analysis of this data.

10. The long sentences are used in the document (line 56-59 and 65-68) it’s better to make simple and small sentence so that the reader may get benefit from the paper.

These sentences have been simplified as below:

The intervention model involved integrating a part-time clinical psychologist into the ward team to facilitate team formulation, reflective practice and train staff to deliver one-to-one and group therapy sessions [10]. Psychologists also delivered psychological therapy on a one-to-one basis to a small case load of service users.

Participating sites were purposely sampled to include a mix of male and female wards in a large metropolitan city, smaller cities, and towns serving more rural populations.

11. The paper doesn't talk much about the specific problems faced during the ethnographic observation part of the study, which could have offered helpful insights into practical difficulties.

This is discussed throughout the paper relating to observers’ awareness/ sensitivity in ward/ group discussion, ward layout, accessible areas e.g., public spaces, transparency of the research/aims of the research (i.e., observations for process evaluation purposes) and building trust/ rapport between the observers and staff/ service user. Specifically, lines 142 – through to 165 and 230 and 238 and interspersed throughout the manuscript.

I have included the following to substantiate points: Lines144-145: Access to areas on the wards was hindered by the ward layout and limited to public spaces potentially reducing privacy for service users.

And lines 149-151: Providing transparency about the research – when and where the observations were taking place – gave service users and staff choice and time to consider whether to participate or not in the observations.

And lines 238-240: This gave the observers an opportunity for building rapport, and trust with service users – becoming part of ‘the group’ rather than being seen as ‘an outsider’.

12. Although the paper extensively covers the importance of personal and team reflective practices, it doesn't go deep into the possible limitations of this approach, missing a critical evaluation of its effectiveness.

We have added the following paragraph in response to this point about the limitations of reflective practices.

Peer debriefing for the researcher focused on their own mental wellbeing is often overlooked in research. It may be overlooked due to lack of time and a failure to make time due to researchers’ potential discomfort in discussing emotions and limited capacity to mentalise. Despite these potential limitations of reflective practice, we viewed researcher debriefing as a fundamental.

13. There's limited exploration of potential biases that might have influenced the researchers' observations and interpretations, which could affect the study's validity and reliability.

We have added the following to supplement information already contained in the paper about the background of the researcher and how members of the team from different disciplines were able to enrich the analysis.

Many of the authors involved in data collection and analysis had experience working in inpatient mental health settings. Researchers therefore already held understandings of ward cultures, processes and the local language used, providing context to interviewees’ experiences. This background influenced interpretations with researchers being more likely to focus on salient features of their own ward experiences, such as wards lacking resources and conflict within staff-patient relationships.

14. The paper doesn't address how the researchers' backgrounds, experiences, and perspectives might have impacted data collection and anal

---

## [Decision Letter · Decision Letter 1]

5 Feb 2025

Conducting Qualitative Research on Acute Mental Health Inpatient Wards: Lessons from the Field

PONE-D-24-24720R1

Dear Dr. Berry,

We’re pleased to inform you that your manuscript has been judged scientifically suitable for publication and will be formally accepted for publication once it meets all outstanding technical requirements.

Kind regards,

Dongmei Li

Academic Editor

PLOS ONE

Additional Editor Comments (optional):

Reviewers' comments:

Reviewer's Responses to Questions

**Comments to the Author**

1. If the authors have adequately addressed your comments raised in a previous round of review and you feel that this manuscript is now acceptable for publication, you may indicate that here to bypass the “Comments to the Author” section, enter your conflict of interest statement in the “Confidential to Editor” section, and submit your "Accept" recommendation.

Reviewer #1: All comments have been addressed

Reviewer #2: All comments have been addressed

2. Is the manuscript technically sound, and do the data support the conclusions?

Reviewer #1: Yes

Reviewer #2: Yes

3. Has the statistical analysis been performed appropriately and rigorously? 

Reviewer #1: Yes

Reviewer #2: (No Response)

4. Have the authors made all data underlying the findings in their manuscript fully available?

Reviewer #1: Yes

Reviewer #2: Yes

5. Is the manuscript presented in an intelligible fashion and written in standard English?

Reviewer #1: Yes

Reviewer #2: Yes

6. Review Comments to the Author

Reviewer #1: Based on the detailed information provided about the manuscript and the authors' revisions in response to the previous feedback, here are my recommendation.

Overall, the authors have made significant strides in addressing the main concerns raised during the initial review. The improvements made to the theoretical grounding, sampling explanation, and ethical considerations greatly strengthen the manuscript.

Given these revisions and their potential contribution to the field, I would suggest that the manuscript be accepted for publication, provided that the authors address any remaining minor concerns, such as adding a brief overview of the findings to give further context to readers.

Reviewer #2: (No Response)

7. PLOS authors have the option to publish the peer review history of their article (what does this mean?). If published, this will include your full peer review and any attached files.

Reviewer #1: No

Reviewer #2: **Yes: **Sadia Mushtaq

---

## [Editor Report · Acceptance letter]

PONE-D-24-24720R1

PLOS ONE

Dear Dr. Berry,

I'm pleased to inform you that your manuscript has been deemed suitable for publication in PLOS ONE. Congratulations! Your manuscript is now being handed over to our production team.

Kind regards,

on behalf of

Dr. Dongmei Li

Academic Editor

PLOS ONE